# Fuel-Efficiency Improvement by Component-Size Optimization in Hybrid Electric Vehicles

**Swapnil Srivastava [1], Sanjay Kumar Maurya [2,\*] and Rajeev Kumar Chauhan [3]**

[1]  Electrical Engineering Department, United College of Engineering and Research, Allahabad 211010, India
[2]  Electrical Engineering Department, GLA University, Mathura 281406, India
[3]  Department of Electrical Engineering, Gyan Ganga Institute of Technology and Sciences, Jabalpur 482003, India
[\*]  Correspondence: sanjay.maurya@gla.ac.in

**Abstract:** Hybrid electric vehicles (HEV) play an important role in sustainable transportation systems. The component size of HEV plays a vital role in the fuel efficiency of vehicles. This paper presents a divided rectangle (DIRECT) method for component sizing of vehicles to ensure better fuel efficiency and satisfying drivability. A state–space model was used to represent the design problem. A constraint multi-input multi-output optimization problem was solved by our DIRECT optimization algorithm. Efficacy of the algorithm was tested with standard drive cycles, including drive cycles for Indian urban and highway conditions representing various driving scenarios in the country. The simulation results illustrated the effectiveness of the proposed algorithm.

**Keywords:** HEV; component sizing; DIRECT optimization; fuel efficiency

## 1. Introduction

Environmental and energy scenario constraints compel us to look for other alternatives to conventional fuel [1]. The automobile is an integral part of social structure and contributes mainly to global warming by emitting a large amount of greenhouse gases (GHG) [2]. Emission from automobiles mainly contain carbon dioxide ($CO_2$), carbon monoxide (CO), nitrogen oxide (NOx), and particulate matter (PM). The exhaust emissions during normal running are far lower than during cold start conditions [3,4]. A lower use of gasoline would limit the emission of health hazardous gasses and make the world a better place. Hybrid electric vehicles (HEV) and pure electric vehicles (EV) are possible alternatives for conventional transportation [5]. The long-term solution would be EV, but due to social acceptance and the high initial cost of EV there is a market for HEV, which may work as a transitional phase between pure ICE and pure EV industry. Hybrid electric vehicles provide an opportunity to minimize fuel-use by combining an electric motor (EM) with a conventional internal combustion engine (ICE) [6].

The literature on energy management of hybrid electric vehicles has a broad spectrum. The complex nature of HEV provides plenty of opportunities for energy management. An energy management system is an algorithm which implements various schemes to increase fuel economy and minimize losses. Rule-based energy management strategies are effective for real-time supervisory control. The rules are usually defined based on user experience, mathematical models, or intuition. The basis of a parallel HEV power train model and the corresponding control laws were developed in [7]. For managing torque distribution, control algorithms were formed as multi-objective non-linear optimization problems; the objective function was then linearized and solved with a charge-sustaining control strategy [8]. The Makov speed interval forecast with a co-evolutionary algorithm was utilized for determination of optimal energy distribution. It dealt with the worst

driving conditions in min–max mode [9]. The decision-making capability of a fuzzy interface system makes it suitable for designing and implementing the basic rules for effective energy management. The nonlinearity and intelligence of fuzzy control make it a powerful tool for energy management of HEV. Fuzzy control is simple, easy to implement, adaptive, and robust in nature. A fuzzy logic-based EMS for an electric–heat hybrid engine power train was developed in [10]. The strategy implemented control of the amount of energy flow between components to satisfy the drive condition, optimize energy efficiency, and minimize pollutants [11]. Fuzzy logic was utilized in suggesting the appropriate size of the components in HEV. Reference power profiles and datasheets were used as inputs for the sizing methodology [12]. The control parameter of an FLC was adjusted using a genetic algorithm in [13]. The proposed scheme considered constraints and targets within a multi-objective optimization function.

Developments in artificial intelligence techniques and the various learning-based techniques make them attractive for energy management of HEV. Deep reinforced learning (DRL) frameworks provide better flexibility for energy management. They can achieve nearly optimal fuel efficiency with a locally trained strategy. Prior knowledge of the route is not necessary to make the method more generalized for a real-world scenario [14]. DRL enables an EMS to learn from the drive cycle; hence it is suitable for online applications of energy management [15]. The model-free approach improves DLR for HEV applications [16]. The fast Q learning approach accelerates the convergence process, thereby decreasing the computational burden. Cloud computation further decreases the computational burden [17]. A DRL algorithm used with an equivalent consumption minimization strategy (ECMS) manages highly complex state–space actions and trades off between real-time applications and learning methodologies [18]. Artificial neural networks (ANN) are other powerful tools used in the energy management application of HEV. ANN provide multi-mode energy management options, which provide the driving pattern required for different driving scenarios [19]. An ANN was used for determining equivalent functions utilized in equivalent consumption minimization strategies and had the capabilities to be trained by real-world data [20]. Equivalent consumption minimization strategy (ECMS) uses an equivalent factor that represents total fuel consumption in terms of actual ICE consumption and equivalent motor consumption. Thus, both electrical and mechanical energy may be represented by a single mathematical term, and management becomes easier. For developing ECMS, the knowledge of future predictions is not necessary and few control parameters are sufficient [21]. A direct search algorithm can minimize a real-valued function as in the case of HEV. In this method, knowledge of the gradient of the objective function is not essential. It can be utilized for solving non-differentiable, noncontinuous objective functions. This algorithm was effectively utilized for the optimization of design and control of HEV [22]. The method is capable of determining global optima of fuel efficiency as an objective function in experimental driving scenarios [23]. Fast responses and high power densities relieve stress on the battery during fast fluctuations. A supercapacitor regenerates energy during braking using adaptive low-pass filter techniques [24]. Dynamic programming (DP) solves the optimization problem by breaking it into small sub-problems and then solving those sub-problems. It is a recursive optimization method. DP determines the optimal torque and gear ratio for a specified drive cycle. Motor rating is determined by the gear ratio and optimal torque [25–27]. A genetic algorithm (GA) is capable of simultaneous optimization of the components and the energy management strategy. Optimization problems have to form an electrical assist control scheme. The complete set of control variables and sizes has to be encoded in chromosomes [28]. A non-dominating shorting GA was used for simultaneous optimization of the power train and EMS, which were not converted into multi-objective functions but used as a single objective function [29]. Particle swarm optimization (PSO) can be used in continuous non-linear functions and is a robust, inexpensive, and fast algorithm [30]. The PSO algorithm was used for solving a fuel efficiency optimization problem expressed as ECMS in [31]. A predictive control method is used for the prediction of future outputs on the

basis of past data of the system. It may also anticipate future inputs. The method focuses on function rather than structure of the model; it produces a mathematical presentation, viz., a transfer function, state equation, or impulse response suitable for its application. The key elements include the predictive model, range of optimization, and feedback correction [32]. A predictive control strategy is capable of application in real-time [33]. The butterfly optimization algorithm (BOA) was used for solving optimization problems based on dynamic-source behavior and power requirements during driving. A multi-objective problem determines the manufacturing cost, running expenses, and associated weight. A decentralized feedback control system for providing optimal velocity trajectory is proposed in [34]. The scheme also provides an effective torque distribution scheme.

Here we provide an algorithm for obtaining best design values for obtaining better fuel economy. The remainder of the paper is as follows: Section 2 provides mathematical modeling of various components for better understanding of dynamic behavior of vehicles during operation. Section 3 covers the optimization problem formulation using a state–space model. Section 4 gives the details of DIRECT optimization techniques. The results and related discussion are covered in Section 5. Finally, Section 6 concludes the findings of the paper.

## 2. Modeling of Hybrid Electric Vehicles

The architecture of HEV is broadly classified on the basis of sequence of energy flow between the components. The basic classifications of HEV are serial and parallel. The serial architecture suffers the drawbacks of double energy conversion processes and associated losses, additional weight of generators, and traction-motor-sizing requirements. The parallel architecture, shown in Figure 1, uses both the IC engine and the motor to propel the drive, and thus provides opportunity to switch the IC engine on and off for controlling the fuel economy [35,36].

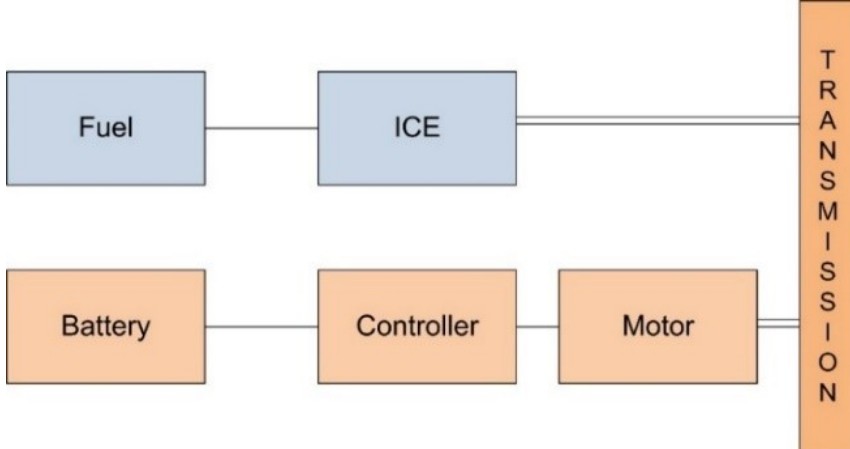

**Figure 1.** Architecture of parallel HEV.

The two inputs of driving force make the modeling of HEV and their components more complex than that of conventional vehicles. Mathematical modeling includes comprehensive, practical, updated, and easy to implement schemes. This section focuses on modeling schemes of HEV and their sub-systems.

### 2.1. ICE Modeling

An ICE converts chemical energy to electrical energy through the burning of fossil fuels, thereby creating a high amount of temperature and pressure. It operates in four operating modes, viz., cranking, idle, ICE OFF, and ICE ON modes.

### 2.1.1. Cranking

The EM drives the ICE in this mode through a negative torque produced. A starter transfers the torque to start the IC engine. The cranking torque is

$$\tau_{crank} = J_{eng} \frac{dW_{eng}}{dt} + \tau_{access} + \tau_{cct} \tag{1}$$

and

$$\omega_{eng} = \frac{1}{J_{eng}} \int_0^t \left( \tau_{crank} - \tau_{access} - \tau_{cct} \right), \tag{2}$$

where $\tau_{access}$ and $\tau_{cct}$ are lumped torque of mechanical accessories and closed throttle torque. The closed throttle torque ($\tau_{cct}$) of ICE is

$$\tau_{cct} = \alpha_1(T)d\delta(t) + \alpha_2(T)sgn(\omega) + \alpha_3(T)\left(\frac{\omega}{\omega_{max\_eng}}\right) + \alpha_4(T)\left(\frac{\omega}{\omega_{max\_eng}}\right)^2. \tag{3}$$

$\alpha_1$, $\alpha_1$, $\alpha_3$, and $\alpha_4$ are coefficients for static friction, Coulomb friction, viscous friction, and air compression torque; $\delta$ is the Dirac delta function; T is the temperature; $\omega$ is the angular speed, and $\omega_{max\_eng}$ is the maximum allowed speed.

### 2.1.2. ICE off

In this mode, the negative torque is provided by the brakes. The driver shaft clutch is to be engaged for running the vehicle. The function of ICE is modeled as

$$\tau_{ref} = \tau_{access} + \tau_{cct} \tag{4}$$

and

$$\tau^* = \tau_{ref} + f(\Delta\omega(t)). \tag{5}$$

The required speed is maintained by the governor mechanism:

$$\tau_{eng\_off} = \tau_{access} + \tau_{cct} + \frac{P_{acc}}{\omega_{eng}} \tag{6}$$

and

$$\omega_{eng} = \omega_{shaft}. \tag{7}$$

### 2.1.3. Idle

In this mode, the ICE clutches are disengaged and the governor maintains the ICE speed. The function is described as

$$\tau_{ref} = \tau_{access} + \tau_{cct}, \tag{8}$$

$$\tau^* = \tau_{ref} + f(\Delta\omega(t)), \tag{9}$$

$$\Delta\omega(t) = \omega_{idle\_desired} - \omega_{idle\_actual}, \tag{10}$$

and

$$\tau_{access} = \frac{P_{access}}{\omega_{ICE}}. \tag{11}$$

2.1.4. ICE on

The clutches are engaged and the propulsion power is provided by the engine and is described as

$$\tau_{access} = \frac{P_{access}}{\omega_{ICE}}.$$ (12)

The closed throttle torque is given by

$$\tau_{cct} = \alpha_1(T)d\delta(t) + \alpha_2(T)sgn(\omega) + \alpha_3(T)\left(\frac{\omega}{\omega_{max\_eng}}\right) + \alpha_4(T)\left(\frac{\omega}{\omega_{max\_eng}}\right)^2.$$ (13)

The torque required for acceleration is

$$\tau_a = J_{ICE}\frac{d\omega}{dt}.$$ (14)

The torque generated by the ICE is

$$\tau_{ICE} = \tau_{demand} + \tau_{access} + \tau_{cct} + \tau_a$$ (15)

with a maximum torque constraint of

$$\tau_{ICE} \leq Max(\tau_{ICE}).$$ (16)

*2.2. Motor Modeling*

In propulsion mode, the motor drives the mechanical load; while in regenerative mode, the SOC of the battery is regained through charging.

2.2.1. Propulsion Mode

In this mode of operation, the motor provides the full or a part of the demanded torque, depending on the EMS strategy. The torque compensates for the inertia and overcomes the losses. The output torque of the motor is

$$\tau_{motor} = \tau_{demand} + \tau_{spin\_loss} + J_{motor}\frac{d\omega}{dt}$$ (17)

with a constraint of maximum torque as

$$\tau_{motor} \leq Max(\tau_{motor}).$$ (18)

The torque lost in spinning is

$$\tau_{spin\_loss} = \alpha_1\delta(t) + \alpha_2\omega + \alpha_3 sgn(\omega).$$ (19)

The electrical power required is

$$P_{electrical} = \frac{P_{mechanical}}{\eta_{motor}}.$$ (20)

The motor-demands for voltage and current are

$$V_{motor} = V_{bus}$$ (21)

and

$$I_{motor} = \frac{P_{electrical}}{V_{bus}},$$ (22)

where $J_{motor}$ is motor inertia, $\tau_{demand}$ is demanded torque, $\tau_{spin\_loss}$ is frictional loss in motor movement, $P_{mechanical}$ is the output mechanical power, $P_{electrical}$ is the input electrical power, $\eta_{motor}$ is motor efficiency, $V_{motor}$ and $I_{motor}$ are voltage and current in the motor, and $V_{bus}$ is the voltage of the DC bus [37].

### 2.2.2. Regenerative Mode

In this mode of operation, the motor behaves as a generator and supplies energy back to the source. The machine provides negative torque through the braking operation. The braking torque is modeled as

$$\tau_{regeneration} = \tau_{demand} - \tau_{spin\_loss} + J_{motor}\frac{d\omega}{dt} \tag{23}$$

with a maximum torque constraint of

$$\tau_{regeneration} \leq Max(\tau_{regeneration}). \tag{24}$$

The electrical power generated is

$$P_{electrical} = \eta_{regeneration}P_{mechanical}. \tag{25}$$

The motor-desired voltage and current are

$$V_{motor} = V_{bus} \tag{26}$$

and

$$I_{regeneration} = \frac{P_{electrical}}{V_{bus}}. \tag{27}$$

### 2.2.3. Spinning Mode

In this mode, the motor is not connected to a mechanical load, it simply moves freely without any mechanical attachment to it. The motor does not drive the load and compensates for spinning loss only, given by

$$\tau_{motor} = \tau_{spin\_loss} = \alpha_1\delta(t) + \alpha_2\omega + \alpha_3\,sgn(\omega). \tag{28}$$

### 2.3. Battery Modeling

The battery provides the electrical power required by the EM for driving the vehicle. The equivalent circuit of the battery is represented in Figure 2.

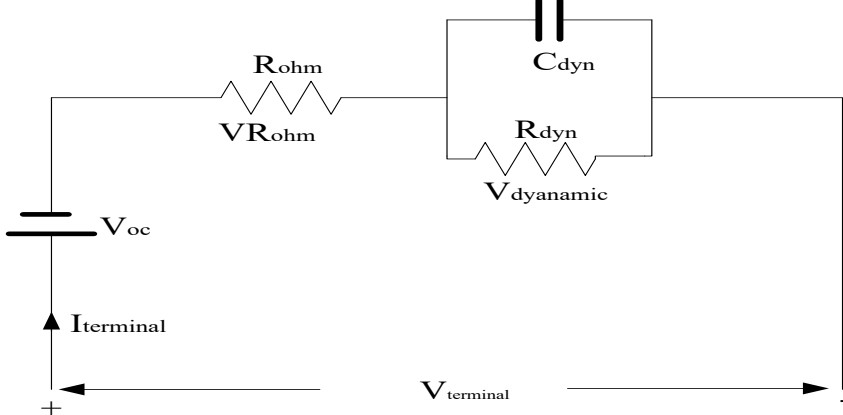

**Figure 2.** Battery model.

The terminal voltage is given by

$$V_{terminal} = V_{oc} + V_{ohm} + V_{dynamic}.$$ (29)

The open circuit voltage is

$$V_{oc} = f(SOC, T).$$ (30)

The ohmic resistance voltage is

$$V_{ohm} = IR_{ohm}(SOC, T).$$ (31)

The dynamic voltage is

$$I_{Rdyn} = \frac{V_{dyn}}{R_{dyn}(SOC, T)}$$ (32)

and

$$I_{Cdyn} = C_{dyn}(SOC, T)\frac{dV_{dyn}}{dt}.$$ (33)

The total current (I) is

$$I = I_{Rdyn} + I_{Cdyn}.$$ (34)

The dynamic voltage in a differential equation form is

$$\frac{dV_{dyn}}{dt} + \frac{dV_{dyn}}{R_{dyn}(SOC, T)} = \frac{I}{C_{dyn}(SOC, T)}.$$ (35)

The overall differential equation for the battery system is

$$\frac{dV_{terminal}}{dt} + \frac{V_{terminal}}{R_{dyn}C_{dyn}} = R_{ohm}\frac{dI}{dt} + \frac{R_{dyn} + R_{ohm}}{R_{dyn}C_{dyn}}I + \frac{V_{oc}}{R_{dyn}C_{dyn}}.$$ (36)

The SOC may be determined by

$$SOC(t) = SOC(t_i) + \frac{1}{Cap_{AHr} \cdot 3600}\int_{t_i}^{t} I(t)\eta_{battery}(SOC, T, sign[I(t)]dt.$$ (37)

The initial SOC is

$$SOC(t_i) = SOC_i$$ (38)

where $V_{terminal}$ is the terminal voltage, $V_{oc}$ is the open circuit voltage of the battery, I is the current, $R_{ohm}$, $R_{dyn}$, and $C_{dyn}$ are the battery parameters, $\eta_{battery}$ is the Coulomb efficiency, and $Cap_{AHr}$ is the capacity in amp hours.

## 3. Problem Formulation

The various modes of energy management are mainly decided on the basis of demanded torque, thus the continuous variable transmission (CVT) position, and torque of ICE and EM are selected as control variables. State of charge, SOC, has a direct impact on energy management schemes of HEV, thus SOC is chosen as one of the state variables. The other state variables are interlocking CVT ($I_{CVT}$), ICE torque, ICE speed, and battery SOC. The optimization problem is formed by a vehicle model in a state-–space model. The vehicle model is formulated as

$$\text{control vector: } u(t) = \begin{pmatrix} u_1 \\ u_2 \\ u_3 \end{pmatrix} = \begin{pmatrix} \delta_{CVT} \\ \tau_{ICEs} \\ \tau_{EMs} \end{pmatrix}, \tag{39}$$

$$\text{state vector: } x(t) = \begin{pmatrix} x_1 \\ x_2 \\ x_3 \\ x_4 \end{pmatrix} = \begin{pmatrix} I_{CVT} \\ \tau_{ICE} \\ SOC_{bat} \\ \omega_{ICE} \end{pmatrix}, \tag{40}$$

$$\text{state equation: } \dot{X}(t) = f(\overline{u}(t), \overline{x}(t)), \tag{41}$$

and

$$\text{constraint: } \dot{X}(t) = f(\overline{u}(t), \overline{x}(t), \dot{\omega}_2(t), \omega_2(t)), \tag{42}$$

where CVT is the continuous variable transmission. Since $\tau_{em} \ll \tau_{ice}$ it is written as $\tau_{em} = \tau_{ems}$ in place of an additional state–space equation. The boundary conditions, time range, state bounds, inequality conditions, and objective function are essential for the problem formulation:

$$\text{time range: } t_0 \le t \le t_{final}, \tag{43}$$

$$\text{state bounds: } \begin{array}{c} \delta_{iCVTmn} \le u(t) \le \delta_{iCVTmx} \\ i_{CVTmn} \le x(t) \le \delta_{CVTmx} \\ T_{EMmn} \le u(t) \le T_{EMmx} \end{array}, \tag{44}$$

$$\text{boundary conditions: } x_3(t_0) = SOC_s, x_3(t_f) = SOC_s, \tag{45}$$

and

$$\text{inequality constraint: } \begin{array}{c} x_1(t) \ge i_{CVTmx} - 5\omega_2^2 \\ x_2(t) \ge T_{ICEmx}(\omega_1) \\ \omega_2(t).x_1(t) \le \omega_{1mx} \end{array}. \tag{46}$$

The time range is determined by the drive cycle [38–40]. The state bounds and control variables are properties of the drive-type category. The optimal problem is to minimize the vehicle fuel consumption for the defined drive cycle. The physical constraints of the ICE, EM, and battery are to be taken into consideration. The objective function would be

$$\text{Min} \left[ C(u) = \int_0^t f(x(t), u(t)t)dt + g(x(t')dt') \right], \tag{47}$$

where 0 and t are the initial and final times of the drive cycle, respectively. X(t) and u(t) are the state and control variable, respectively, $f(x(t), u(t))$ is the instantaneous fuel consumption, and $g(x(t'))$ is the function for the operating parameters of vehicle. The state has two values:

$$x(t) = \begin{cases} x_1(t): \text{The\_battery\_SC} \\ x_2(t): \text{The\_ICE\_State} \end{cases}. \tag{48}$$

$X_1(t)$ is the continuous state controlled by $\dot{x}_1(t) = f_1(u_1(t),t)$. The value of the second term $X_2(t)$) is either 0 or 1, depending on the ICE's on–off condition. The control variable has three states for optimization:

$$u = \begin{cases} u_1 : Power\_Split\_Ratio \\ u_2 : Gear\_Scheduling..... . \\ u_3 : State\_of\_ICE......... \end{cases} \quad (49)$$

$U_1$ is the classical control method of HEV and $U_3$ is known from the X2(t) state. The fuel consumption is determined by

$$f((x(t), u(t), t) = \begin{cases} f_1(u_1(t), t) : if\_Engine\_ON \\ 0 : If\_Engine\_OFF.............. \\ f_2 : If\_Engine\_Startup......... \end{cases} \quad (50)$$

and

$$g(x(t)) = \frac{K_1}{Fuel\_Economy} + \frac{K_2}{Emmision} + K_3.Mass + \frac{K_4}{Maximum\_Speed} + \frac{K_5}{Gradeability} + \frac{K_6}{Zone\_Acceleration}, \quad (51)$$

where $f_1(u_1(t),t)$ is determined by engine map data and $f_2$ is additional fuel consumed during the start-up process.

## 4. Energy Management System

When the parallel electric-assist control strategy is used, the motor provides additional power per the requirements while it maintains the battery SOC. The variables used for describing the control strategy are summarized in Table 1.

**Table 1.** Variables used in describing the control strategy.

| Variable | Description |
| --- | --- |
| cs_hi_soc | Highest target battery SOC |
| cs_lo_soc | Lowest target battery SOC |
| cs_electric_launch_spd_hi | Vehicle speed below which pure EV mode is switched on |
| cs_electric_launch_spd_lo | Lowest vehicle speed in pure EV mode |
| cs_off_trq_frac | Minimum torque threshold; when controlled at a lower torque, the ICE will switch off if SOC > cs_lo_soc |
| cs_min_trq_frac | Minimum torque threshold; when controlled at a lower torque, the ICE operates at the threshold torque and the motor works in regenerating mode if the SOC < cs_lo_soc |
| cs_charge_trq | Additional torque required by the ICE for recharging the battery when the ICE is ON |

The speed and torque load for the drive condition are fed to the ICE through the clutch assembly. The energy management scheme, shown in Figure 3, decides the torque distribution in the ICE and motor which together produce the required torque while ensuring the battery SOC remains in permissible limits. The cost function is defined as depicted in Figure 3.

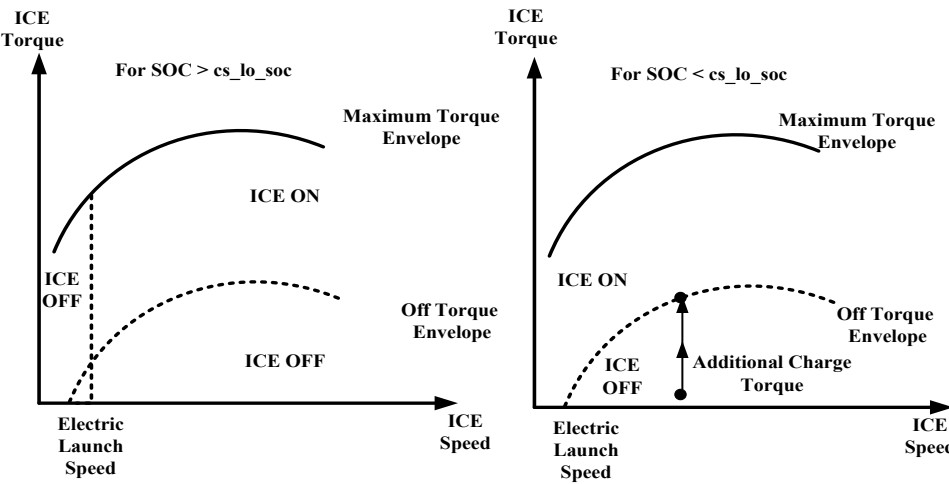

**Figure 3.** Energy management scheme.

When the SOC of the battery is below cs_soc_lo, an additional torque is demanded from the ICE, for charging the battery. This additional torque, required for charging the battery, is proportional to the difference between battery SOC and the average of cs_hi_soc and cs_lo_soc. This prevents the ICE torque from falling below a threshold fraction, cs_min_trq_frac, of the maximum ICE torque for the current speed of the vehicle. This is done to prevent the ICE from operating at an inefficient operation point. The ICE torque is required only when the ICE is on [41,42].

If the speed required is below the electric launch speed, cs_electric_launch_spd, the ICE should be turned off. If the battery SOC is above its lower limit, the ICE should be turned off. If both the requested speed below the launch speed and the battery SOC are above the lower limit, the ICE should be turned off.

If the torque demanded is below the cutoff torque fraction of the maximum torque, cs_off_trq_frac, the ICE should be turned off. If the demanded torque is lower than the cutoff and the battery SOC is higher than the lower limit, the ICE should be turned off. The emissions model is developed using absolute spark advance (SA) as

$$SA = \Delta_{SA} + SA_{optim}. \tag{52}$$

The emissions are determined using the following relations [43]:

$$[CO] = co_1 \left( \theta - 1 + \sqrt{(\theta-1)^2 + co_2} \right), \tag{53}$$

$$[HC] = \max \left( 0, hc_1.N + hc_2.(\theta - 0.9)^2 + hc_3.SA + hc_4.P_{intake} + hc_5 \right). \tag{54}$$

and

$$[NO] = \max \left( [NO]_{min}, no_1.(\theta - 0.9)^2 + no_2.SA + no_3.P_{intake} + no_4 \right), \tag{55}$$

where $\theta$ is the air-to-fuel intake ration, N is the speed in rpm, $P_{intake}$ is the input power, and $co_x$, $no_x$, and $hc_x$ are adjusting parameters determined in experiments in [44].

## 5. Optimization Process

To solve the optimization problem, a divided rectangle (DIRECT) optimization technique is used. Developed by Donald R. Jones, it is a sampling-based technique and is a modified version of the Lipschitzian method developed by Jones et al. in 1993 [45,46]. The DIRECT algorithm eliminates use of the Lipschitz constant by identifying all possible values and using a balanced approach for local and global search. The algorithm starts by scaling of a design box in an n-dimensional hypercube. The search is initiated by

evaluation of an objective function at the center of the hypercube. The algorithm divides the probable optimal rectangle via sampling of the longest coordinate direction of the rectangle, as shown in Figure 4. Sampling is done in such a manner that a sampled point becomes the center of an n-dimensional rectangle. The division is continued until the termination criterion is achieved.

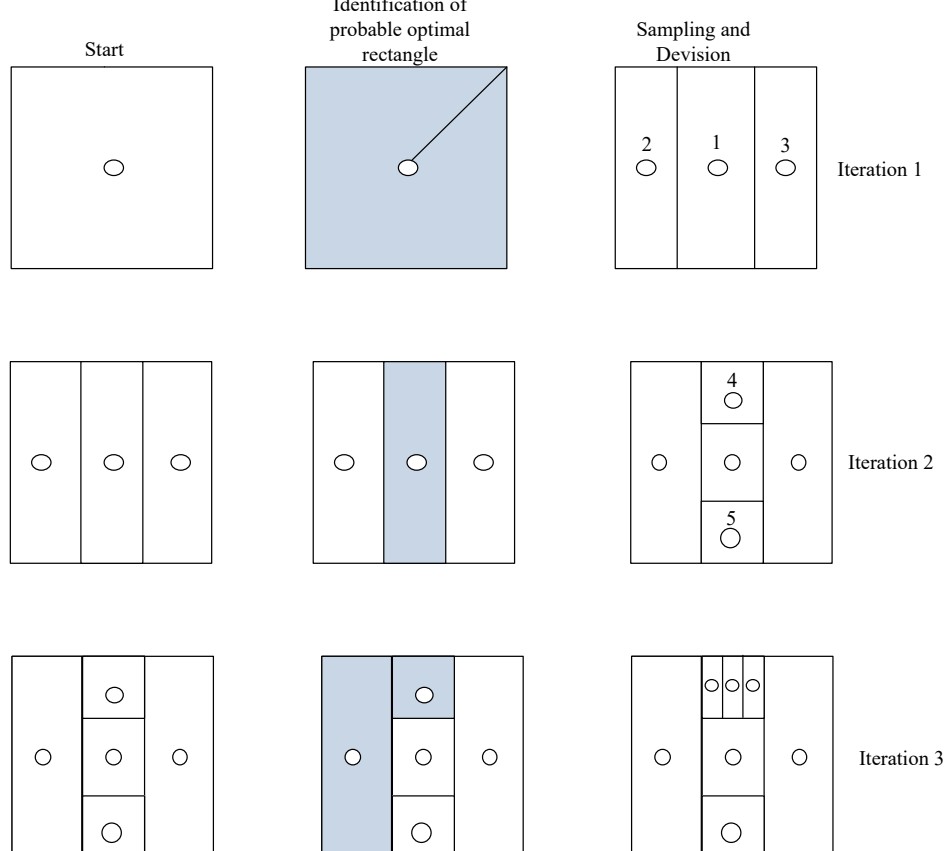

**Figure 4.** Hyper-rectangle formation in the DIRECT algorithm.

The key factor of DIRECT optimization is the determination of hyper-rectangles as samples. If all the hyper-rectangles are sampled, the search fails to focus on a local optimum and gets stuck on the global optimum. To avoid it, sampling is done on hyper-rectangles that are probably optimal. In Figure 5a, the plot between the objective function f $(c_i)$ and the size of a rectangle $(d_i)$ is given for all hyper-rectangles, $i \in R$, shown by dots. The vertical axis provides better results for the local search as the lower values of $f(c_i)$ are close to the actual global minima. The x-axis provides better results for the global search as larger di values indicate higher unexplored zones and have more chances of improvement [47,48].

Let $\Gamma$s be the set of all possible hyper-rectangles with size d, s = 1, 2,..., S. Therefore, each $\Gamma$s contains all possible hyper-rectangles (i) with the same dashed boxes as shown in Figure 5b, i = 1, 2,...,. Let $\Phi(\Gamma$s) be the hyper-rectangle with the best objective function value in $\Gamma$s, thus

$$\phi(\Gamma s) = \left\{ i : f(c_i) \le f(c_j), i, j \in I \right\}. \tag{56}$$

Shown in Figure 5c, let € be the set of all best-objective-function hyper-rectangles ($\Phi(\Gamma$s)), then

$$€ = U\ (\Phi(\Gamma s)). \tag{57}$$

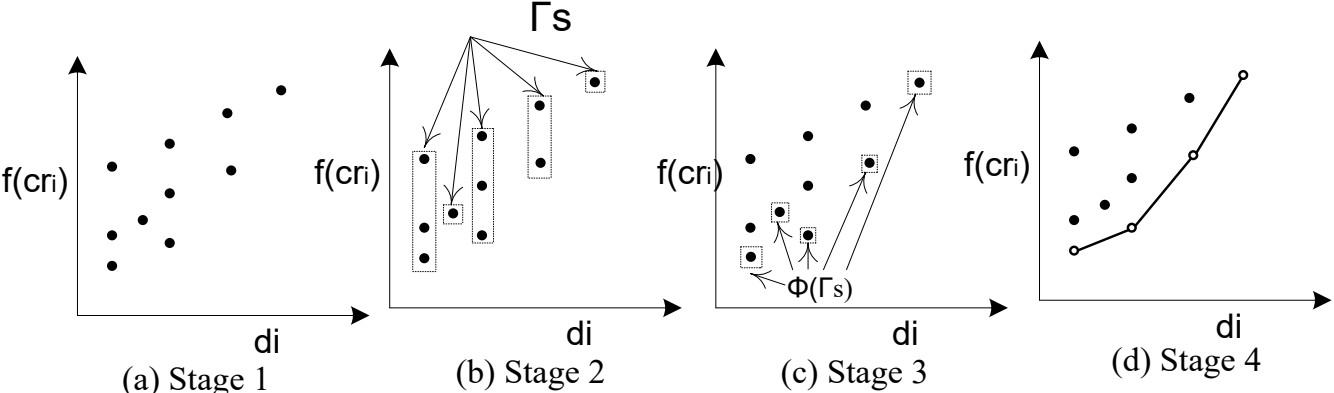

**Figure 5.** Identification of hyper-rectangles.

The rate-of-change parameter, k, defines the scope of the search between local and global optimum. The 'k' value changes during the process of optimization for identifying a set U of probable optimal-hyper-rectangles. A hyper-rectangle (i) is said to be a probable optimal-hyper-rectangle if there exists a value of k > 0 so as

$$f(cr_i) - kd_i \le f(cr_j) - kd_j; j = 1, 2, .... , \tag{58}$$

$$f(cr_i) - kd_i \le f(cr^*) - e \left| f(cr^*) \right|, \tag{59}$$

where cr and $f(cr^*)$ are center- and objective-function value of the best hyper-rectangle, respectively, and e is a small positive constant. Equation (58) ensures that hyper-rectangles depicted as lower right dots in Figure 5d are probable optimal points, while Equation (59) defines the lower bounds ($f(cr_i-kd_i)$) of every probable optimal-hyper-rectangle below the lower limit of the current solution, thereby preventing the search process from becoming too local-optima centric.

The algorithm determines the set U in each iteration, samples each i ∈ U, and then divides each i ∈ U into smaller hyper-rectangles. Each hyper-rectangle becoming smaller and the sampling process depending on dimension size, avoids repetition of samples in the solution space. The iteration stops at fulfilling the function-evaluation criteria. The initial conditions, lower bounds, and upper bounds of design variables are given in Table 2.

**Table 2.** Design variable parameters.

| Variable Name | Initial Condition | Lower Bound | Upper Bound |
|---|---|---|---|
| cs_lo_SOC | 0.4 | 0.1 | 0.5 |
| cs_hi_SOC | 0.8 | 0.55 | 1 |
| cs_charge_trq | 15.25 | 1 | 80.9 |
| cs_min_trq_frac | 0.4 | 0.05 | 1 |
| cs_off_trq_frac | 0.05 | 0.05 | 1 |
| cs_electric_launch_spd_lo | 0 | 0 | 15 |
| cs_electric_launch_spd_hi | 10 | 10 | 30 |

Figures 6 and 7 show the design iteration and design variable for achieving the objective function before and after the optimization.

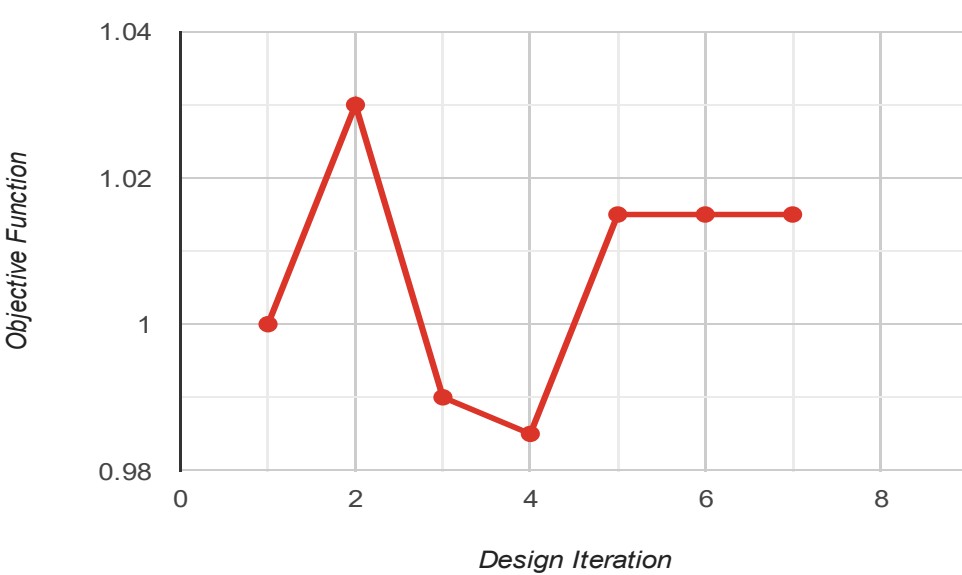

**Figure 6.** Design iterations.

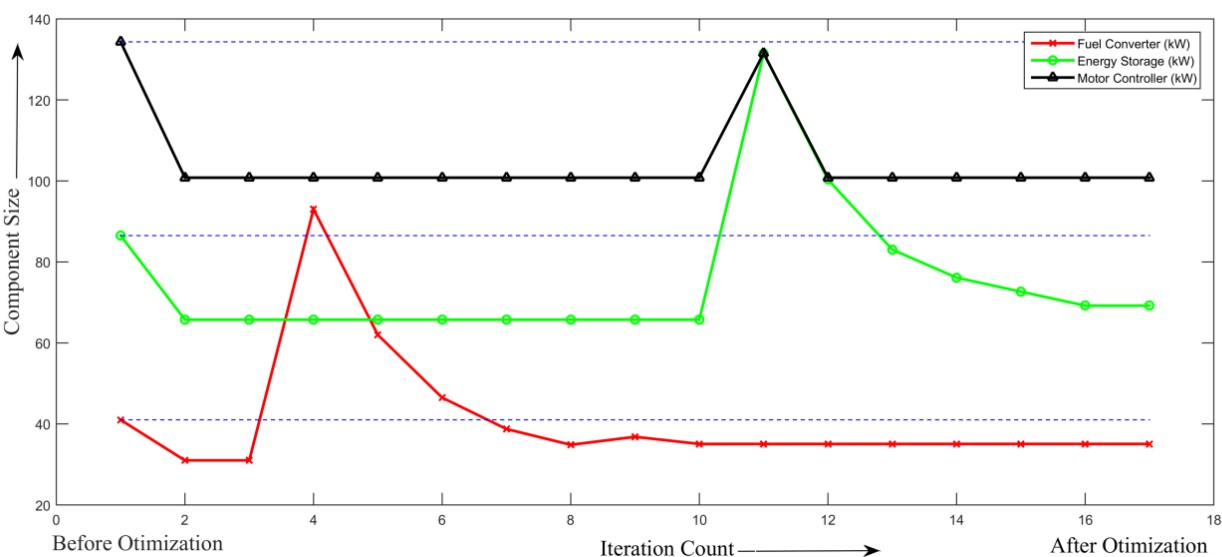

**Figure 7.** Design variable before and after optimization.

The grade and acceleration constraints with tolerance limits used in this study are given in Tables 3 and 4, respectively.

**Table 3.** Grade constraints.

| Parameter | Goal | Tolerance |
|---|---|---|
| Speed (mph) | 55 | 0.01 |
| Grade (%) | 6 | 0.05 |

**Table 4.** Acceleration constraints.

| Speed Range | Goal | Tolerance |
|---|---|---|
| 0–18 mph | 3.5 | 0.02 |
| 0–30 mph | 10 | 0.02 |
| 0–60 mph | 12 | 0.02 |
| 40–60 mph | 5.3 | 0.02 |
| 0–85 mph | 23.4 | 0.05 |

## 6. Results and Discussion

The performance of optimized vehicles was checked in advanced vehicle simulator (Advisor) run in MATLAB environment. Various standard cycles were used for checking the efficacy of the algorithm. The parameters chosen for the test vehicle are summarized in Table 5.

**Table 5.** Vehicle parameters.

| Parameter | Value |
|---|---|
| Drive train | Parallel |
| ICE type | SI |
| Battery | Lead acid (308 V) |
| Motor | AC_75 |
| Transmission | Manual |
| Vehicle mass | 1350 kg |

A fuel efficiency comparison after the optimization process for standard drive cycles, including two drive cycles in an Indian scenario, is given in Table 6.

**Table 6.** Fuel efficiency comparison (MPG).

| Drive Cycle | Before Optimization | After Optimization | Percent Increase |
|---|---|---|---|
| CYC_UDDS | 42.1 | 75.7 | 79.8 |
| CYC_US06 | 36.6 | 39.1 | 6.8 |
| CYC_FTP | 41.9 | 71.5 | 70.6 |
| CYC_INRETS | 36.5 | 44.2 | 21.1 |
| CYC_WVYSUB | 42.5 | 56 | 31.8 |
| CYC_REP05 | 39.7 | 44.9 | 13.1 |
| CYC_OCC | 31.7 | 76.3 | 140.7 |
| CYC_LA92 | 36.6 | 48.6 | 32.8 |
| CYC_ARB02 | 35.8 | 40.4 | 12.8 |
| CYC_India_Hwy | 49.2 | 57 | 15.9 |
| CYC_India_Urban | 33.8 | 93.2 | 175.7 |

A comparison of the average fuel consumed (MPG) in test drive cycles is given in Figure 8.

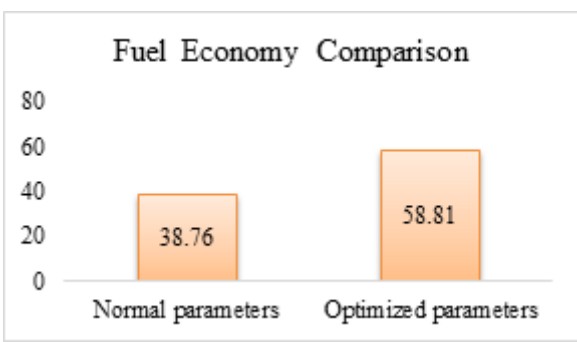

**Figure 8.** Fuel efficiency comparison.

Figure 8 shows the average fuel efficiency, when a vehicle is run with the different drive cycles mentioned in Table 6 as input, before and after optimization of the vehicle parameters. From Figure 8 it can be seen that the optimization process indicates an average increase of 51.73% in fuel efficiency providing substantial savings in fuel cost. The pollutant emissions during vehicle runs, HC, CO, and NOx were measured during the process of testing. The outcomes for pollutant emissions with standard drive cycles [49] are summarized in Table 7.

**Table 7.** Pollutant emission comparison.

| Drive Cycle | Pollutant | Before Optimization | After Optimization | Percentage Change |
|---|---|---|---|---|
| CYC_UDDS | HC | 0.553 | 0.476 | 13.9 |
| | CO | 2.445 | 2.537 | −3.76 |
| | NOx | 0.41 | 0.358 | 12.7 |
| CYC_US06 | HC | 0.54 | 0.505 | 6.5 |
| | CO | 9.07 | 9.625 | −6.1 |
| | NOx | 0.476 | 0.437 | 8.2 |
| CYC_FTP | HC | 0.416 | 0.372 | 106 |
| | CO | 1.978 | 2.155 | −8.9 |
| | NOx | 0.347 | 0.313 | 9.8 |
| CYC_INRETS | HC | 0.355 | 0.324 | 8.7 |
| | CO | 2.304 | 2.452 | −6.4 |
| | NOx | 0.316 | 0.292 | 7.6 |
| CYC_WVYSUB | HC | 0.568 | 0.546 | 3.9 |
| | CO | 2.155 | 2.311 | −7.2 |
| | NOx | 0.386 | 0.392 | −1.5 |
| CYC_REP05 | HC | 0.305 | 0.282 | 7.5 |
| | CO | 4.97 | 5.234 | −5.3 |
| | NOx | 0.295 | 0.265 | 10.2 |
| CYC_OCC | HC | 0.67 | 0.593 | 11.5 |
| | CO | 2.802 | 2.547 | 9.1 |
| | NOx | 0.428 | 0.394 | 7.9 |
| CYC_LA92 | HC | 0.473 | 0.447 | 5.5 |
| | CO | 3.166 | 4.461 | −40.9 |
| | NOx | 0.416 | 0.397 | 4.6 |
| CYC_ARB02 | HC | 0.325 | 0.291 | 10.4 |
| | CO | 5.51 | 5.317 | 3.5 |
| | NOx | 0.324 | 0.303 | 6.5 |
| CYC_India_Urban | HC | 0.477 | 0.399 | 16.4 |

|  |  |  |  |  |
| --- | --- | --- | --- | --- |
|  | CO | 2.15 | 1.846 | 14.1 |
|  | NOx | 0.377 | 0.317 | 15.9 |
| CYC_India_Hwy | HC | 0.582 | 0.573 | 1.5 |
|  | CO | 2.457 | 2.416 | 1.7 |
|  | NOx | 0.447 | 0.501 | −12.1 |

The average values of pollutant emission for different standard drive cycles were calculated. Comparative average results of various pollutant emissions are shown in Figure 9. It can be seen that the optimization process reduces the average HC and NOx emission by 8.33% and 5.26%, respectively.

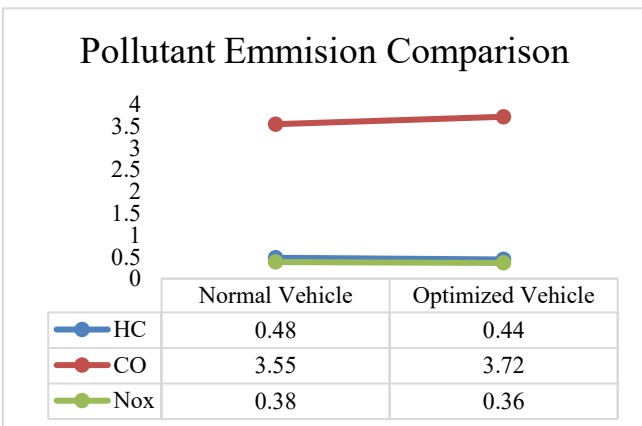

**Figure 9.** Pollutant emission comparison.

Figures 10 and 11 represent the power consumed by each component of an HEV. An analysis of Figures 10 and 11 indicates that the power consumed by the ICE is substantially reduced after the optimization process. The impact of optimization is a better fuel efficiency obtained during vehicle runs.

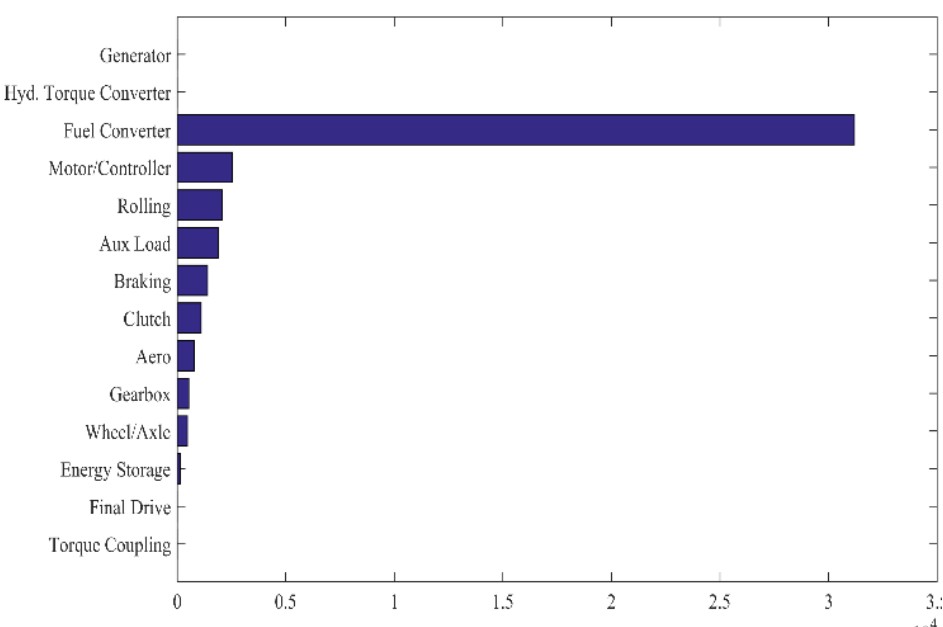

**Figure 10.** Energy usage (kJ) of HEV before optimization.

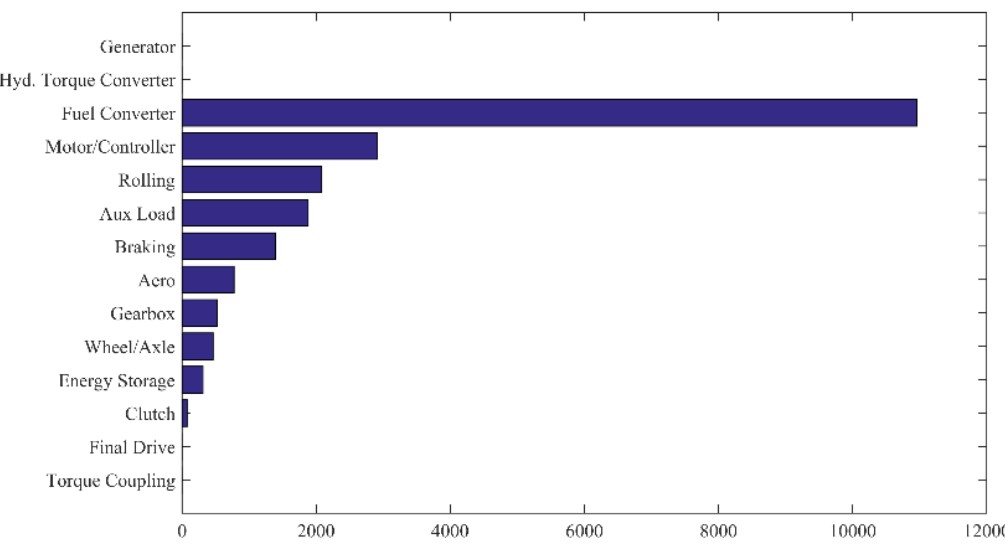

**Figure 11.** Energy usage (kJ) of HEV after optimization.

The SOC and pollutant emissions for unoptimized and optimized vehicles are shown in Figures 12 and 13, respectively. The SOC profile indicates a suitable zone of operation for longer battery life. The pollutant emissions decrease after implementation of the optimization algorithm.

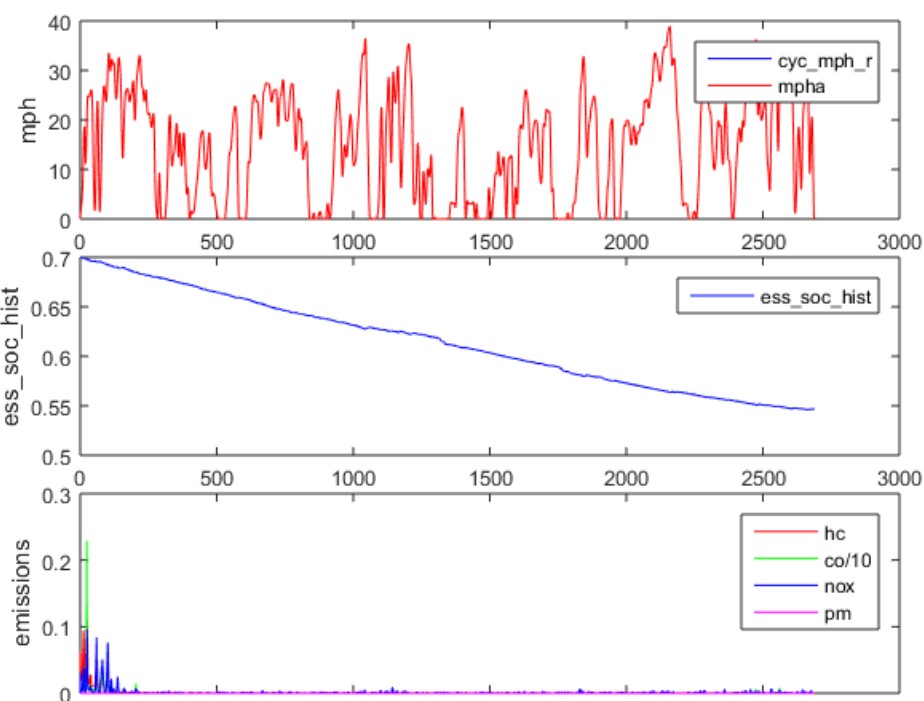

**Figure 12.** SOC and emissions before optimization.

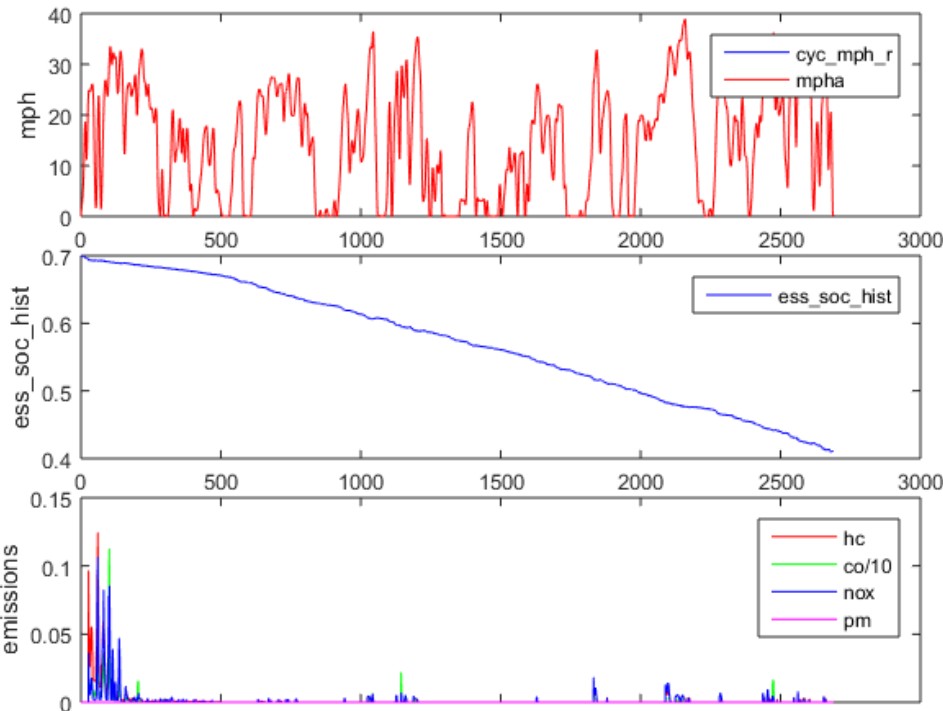

**Figure 13.** SOC and emissions after optimization.

## 7. Conclusions

The paper presents a DIRECT optimization algorithm for component sizing of HEV. The proposed method was tested for various standard drive cycles including Indian-scenario drive cycles. The proposed algorithm provided an average increase of 51.73% in fuel economy; additionally, the emission of the pollutant components HC and NOx were reduced by 8.33% and 5.26% on average, respectively. The method reduced the energy consumed by the ICE itself by 64.1%, showing a great overall improvement in fuel economy. The algorithm indicated optimal SOC of the battery during the operation. Simulation results indicate the effectiveness of the proposed optimization algorithm. Future work may include the effect of battery chemicals and thermal properties in the analysis.

**Author Contributions:** S.S. developed the theoretical formalization, and performed the mathematical analysis and the analytic calculations. He also developed the project in MATLAB and Simulink software. S.K.M. was involved in implementation of the optimization algorithm and in the write-up of the paper. All authors have read and agreed to the published version of the manuscript.

**Funding:** This research received no external funding.

**Data Availability Statement:** Not applicable.

**Conflicts of Interest:** The authors declare no conflicts of interest.

## Abbreviations

| Symbol/Abbreviation | Meaning |
| --- | --- |
| T | Torque (Nm) |
| $\tau_{crank}$ | Cranking torque (Nm) |
| $\tau_{access}$ | Lumped torque of mechanical accessories (Nm) |
| $\tau_{cct}$ | Closed throttle torque (Nm) |
| $J_{eng}$, $J_{motor}$ | Engine and motor inertia (kg) |
| $\alpha_1$, $\alpha_2$, $\alpha_3$, $\alpha_4$ | Coefficients for static friction, Coulomb friction, viscous friction, and air compression torque |
| $\omega$ | Angular speed (rad/sec) |

| | |
|---|---|
| $\tau_{ref}$, $\tau_{demand}$ | Reference and demanded torque (N-m) |
| $\tau_{motor}$, $\tau_{regeneration}$ | Motor and regenerative torque (N-m) |
| $\tau_{spin\_loss}$ | Toque loss in spinning (N-m) |
| $P_{electrical}$, $P_{mechanical}$ | Electrical and mechanical power (kW) |
| $V_{bus}$, $V_{terminal}$ | DC bus and terminal voltage (V) |
| I | Current (A) |
| R | Resistance (Ohm) |
| SOC | State of charge |
| T | Temperature (0C) |
| C | Capacitance (F) |
| Dyn | Dynamic |
| AHr | Ampere hour |
| η | Efficiency |
| CVT | Continuous variable transmission |
| x | State variable |
| u | Control variable |
| Γ | Set of possible hyper-rectangles |
| d | Size of rectangle |
| Φ(Γs) | Hyper-rectangle with best-objective-function value |
| € | Set of all best-objective-function hyper-rectangles |
| cr | Center of objective function |

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
