# Peer review of "Fuel-Efficiency Improvement by Component-Size Optimization in Hybrid Electric Vehicles"

_wevj, doi:10.3390/wevj14010024_

Round 1
Reviewer 1 Report
In this manuscript, the authors build a simulation model of a hybrid electric vehicle. Then, the DIRECT method is used to optimize the parameter in the model, and a significant efficiency improvement is achieved. The results are good, but there are some problems, which must be solved before it is considered for publication.
1. Error in subtitle, “2.1.1” and “2.1.3” are both “Cranking”
2. The symbols in the article are confusing. For example, in equation(39) and equation(40), the definition of (u1, u2, u3) is different. In appendix A, the symbol ‘T’ means temperature, but in equation(39) and equation(40), the ‘T’ is not temperature. In equation(40), the meaning of symbol ‘W’ is unclear and ‘W’ is not displayed in the appendix A. The authors should proofread carefully to ensure that each symbol has a clear and unique meaning. The current confusion makes it hard to understand this manuscript.
3. In this manuscript, many subscripts are used, and the meaning of these subscripts should be explained. An additional table is necessary. For example, I don’t know the meaning of ‘CVTs’ in equation(39). What is the difference between ‘CVTs’ and ‘CVT’?
4. In Table 1, the variable names are not defined. What is the meaning? What is the relationship between these variables and those equations?
5. The main work of this manuscript is the model, and many parameters are used. It is necessary to give specific values of parameters so that other authors can evaluate the rationality of the result.
6. The values of parameters before/after the optimization should be shown.
7. The energy management formulas of the hybrid power system are not shown. They are very important for the efficiency of the vehicle.
8. The authors show the pollutant results, but there are no equations about the pollutant.
9. When considering the fuel consumption of a hybrid vehicle, it is necessary to consider the difference between the battery energy of the initial state and the final state, and how to compensate this energy into the fuel consumption. In figure 10 and figure 11, the SOC is different, so the consumption of battery energy is different, and these two results cannot be compared directly. The authors don’t elaborate on this problem.
10. In line 257, there seems to be a paragraph missing. Where are the formulas of the vehicle model?
In summary, the authors build a huge hybrid vehicle model and consider the impact of multiple factors on energy consumption. This work is meaningful. However, as a research paper, the authors do not present all the details of the model rigorously and carefully. The authors’ casual writing is unacceptable. It is hard to read this manuscript because of the confusion in symbols and presentation.
Author Response
To, Date: 01.12.2022
Editor-in-Chief
World Electric Vehicle Journal (WEVJ)
MDPI
St. Alban-Anlage 66, 4052 Basel, Switzerland
Reference:
Manuscript Reference No.: wevj-2005928
Submission Title: Fuel Efficiency Improvement by Component Size Optimization in Hybrid
Electric Vehicle
Manuscript Type: Research Article
Date of Submission: 17.10.2022
Subjects: Responses to Reviewers Comments for the Manuscript Reference No.: wevj-2005928
Dear Sir/Madam,
In view of above-mentioned reference, we express our sincere gratitude and thanks to Editor-in-Chief for accepting manuscript for submission and review process, we also express our gratitude and thanks to the eminent reviewers of the submitted manuscript for their review comments to revise the submitted manuscript for the next phase of publication.
Hereby we are submitting the responses of each review comments received through eminent reviewers. Appropriate changes related to review comments are incorporated in the revised manuscript. The responses as authors’ comments are mentioned below in the rebuttal table as follows.
Reviewer’s comments: |
Authors’ comments |
|
Review point: 1. |
Error in subtitle, “2.1.1” and “2.1.3” are both “Cranking” |
Apologies for mistake, the correction is been done in revised manuscript. |
Review point: 2. |
The symbols in the article are confusing. For example, in equation(39) and equation(40), the definition of (u1, u2, u3) is different. In appendix A, the symbol ‘T’ means temperature, but in equation(39) and equation(40), the ‘T’ is not temperature. In equation(40), the meaning of symbol ‘W’ is unclear and ‘W’ is not displayed in the appendix A. The authors should proofread carefully to ensure that each symbol has a clear and unique meaning. The current confusion makes it hard to understand this manuscript. |
Apologies for mistake, the The symbols are corrected in revised manuscript. The proof reading of the manuscript is been done. |
Review point: 3. |
In this manuscript, many subscripts are used, and the meaning of these subscripts should be explained. An additional table is necessary. For example, I don’t know the meaning of ‘CVTs’ in equation(39). What is the difference between ‘CVTs’ and ‘CVT’? |
The required correction is been done in revised manuscript |
Review point: 4. |
In Table 1, the variable names are not defined. What is the meaning? What is the relationship between these variables and those equations? |
The meaning and relevance of variables are defined in newly added section IV. |
Review point: 5. |
The main work of this manuscript is the model, and many parameters are used. It is necessary to give specific values of parameters so that other authors can evaluate the rationality of the result. |
Specific values of design parameters are mentioned in section IV. |
Review point: 6. |
The values of parameters before/after the optimization should be shown. |
The values of the parameters before/after the optimization are mentioned in figure (7) |
Review point: 7. |
The energy management formulas of the hybrid power system are not shown. They are very important for the efficiency of the vehicle. |
Section IV describes the details of energy management scheme |
Review point: 8. |
The authors show the pollutant results, but there are no equations about the pollutant. |
The mathematical formulation of pollutant calculations are covered in section IV |
Review point: 9. |
When considering the fuel consumption of a hybrid vehicle, it is necessary to consider the difference between the battery energy of the initial state and the final state, and how to compensate this energy into the fuel consumption. In figure 10 and figure 11, the SOC is different, so the consumption of battery energy is different, and these two results cannot be compared directly. The authors don’t elaborate on this problem. |
The author has compared the result to show the performance of the vehicle after optimization |
Review point: 10. |
In line 257, there seems to be a paragraph missing. Where are the formulas of the vehicle model? In summary, the authors build a huge hybrid vehicle model and consider the impact of multiple factors on energy consumption. This work is meaningful. However, as a research paper, the authors do not present all the details of the model rigorously and carefully. The authors’ casual writing is unacceptable. It is hard to read this manuscript because of the confusion in symbols and presentation. |
The required correction is been done in revised manuscript, The vehicle model is mentioned in section II. |
The revised manuscript is also uploaded on Journal Management System portal.
Thanks & Regards
First Author:
Swapnil Srivastava
Department of Electrical Engineering
United College of Engineering and Research, Prayagraj,UP, India
Email id: [email protected]
Mobile/Phone: +91-8299652473, 9455419395
Corresponding author:
First Co-Author:
Sanjay Kumar Maurya, Ph.D.
SMIEEE (U.S.A.), MIET (U.K.)
Department of Electrical Engineering
GLA University, Mathura, UP, India
Email id: [email protected]
Phone No. +91-5446250855
Mobile-+91-7895010874
Reviewer 2 Report
The component sizing of HEV is optimized by DIRECT method. The average fuel economy is increased by 51.73% with the decreasing of emission pollutant. However, there are some issues should be handled before it gets published. Major revision is required.
1. The literature review should be improved, the authors just illustrated some current works without identifying the gap between their work and the others’.
2. The percentage of the papers published within 5 five years is less than 25%. For the introduction part, more works in recent years should be added.
3. All the symbols in the equations should be introduced. The layout of the equations should be improved, for example, the footnote of should be smaller than the normal words.
4. Figure 5 should be improved.
5. For the optimization issue, the value of variables before and after optimization is quite important, it should be demonstrated.
6. There are quite a lot drive cycles are initiated. The detail of them could be ignored, but the regulation of the cycles should be cited.
7. The pollutant of the emission is compared, however, the calculation process of them are ignored, which shouldn’t.
8. The label of the x axis in Figures 10 and 11 are missing.
9. The language should be improved, such as, the sentence in line 256 and 257 is incomplete.
Author Response
To, Date: 01.12.2022
Editor-in-Chief
World Electric Vehicle Journal (WEVJ)
MDPI
St. Alban-Anlage 66, 4052 Basel, Switzerland
Reference:
Manuscript Reference No.: wevj-2005928
Submission Title: Fuel Efficiency Improvement by Component Size Optimization in Hybrid
Electric Vehicle
Manuscript Type: Research Article
Date of Submission: 17.10.2022
Subjects: Responses to Reviewers Comments for the Manuscript Reference No.: wevj-2005928
Dear Sir/Madam,
In view of above-mentioned reference, we express our sincere gratitude and thanks to Editor-in-Chief for accepting manuscript for submission and review process, we also express our gratitude and thanks to the eminent reviewers of the submitted manuscript for their review comments to revise the submitted manuscript for the next phase of publication.
Hereby we are submitting the responses of each review comments received through eminent reviewers. Appropriate changes related to review comments are incorporated in the revised manuscript. The responses as authors’ comments are mentioned below in the rebuttal table as follows.
Reviewer’s comments: |
Authors’ comments |
|
Review point: 1. |
The literature review should be improved, the authors just illustrated some current works without identifying the gap between their work and the others’. |
The literature review is upated with latest work in the area, The newly added references in literature review are [9], [12], [13], [16], [17], [18],[19], [20], [24] and [34] |
Review point: 2. |
The percentage of the papers published within 5 five years is less than 25%. For the introduction part, more works in recent years should be added. |
The newly referred papers, [9], [12], [13], [16], [17], [18],[19], [20], [24], [34], [42] and [43], in revised manuscript are from past 3 years. |
Review point: 3. |
All the symbols in the equations should be introduced. The layout of the equations should be improved, for example, the footnote of should be smaller than the normal words. |
The required corrections are made in revised manuscript |
Review point: 4. |
Figure 5 should be improved. |
Quality of figure is improved in revised manuscript |
Review point: 5. |
For the optimization issue, the value of variables before and after optimization is quite important, it should be demonstrated. |
The values of the parameters before/after the optimization are mentioned in figure (7) |
Review point: 6. |
There are quite a lot drive cycles are initiated. The detail of them could be ignored, but the regulation of the cycles should be cited. |
The regulation of drive cycles are cited as [49] in revised manuscript |
Review point: 7. |
The pollutant of the emission is compared, however, the calculation process of them are ignored, which shouldn’t. |
The mathematical formulation of pollutant calculations are covered in newly added section IV |
Review point: 8. |
The label of the x axis in Figures 10 and 11 are missing. |
The label of the x axis in Figures are mentioned in revised manuscript |
Review point: 9. |
The language should be improved, such as, the sentence in line 256 and 257 is incomplete. |
The required corrections are made in revised manuscript |
The revised manuscript is also uploaded on Journal Management System portal.
Thanks & Regards
First Author:
Swapnil Srivastava
Department of Electrical Engineering
United College of Engineering and Research, Prayagraj,UP, India
Email id: [email protected]
Mobile/Phone: +91-8299652473, 9455419395
Corresponding author:
First Co-Author:
Sanjay Kumar Maurya, Ph.D.
SMIEEE (U.S.A.), MIET (U.K.)
Department of Electrical Engineering
GLA University, Mathura, UP, India
Email id: [email protected]
Phone No. +91-5446250855
Mobile-+91-7895010874
Reviewer 3 Report
The paper addresses the problem of adequately designing the size of hybrid vehicle components in order to ensure better fuel efficiency and satisfying drivability conditions. The topic is exciting, but the reviewer has significant concerns:
1) the article is not easy to read and needs a major English-style update;
2) the state of the art needs to be enhanced. For instance see ([Wu, X., Cao, B., Li, X., Xu, J., & Ren, X. (2011). Component sizing optimization of plug-in hybrid electric vehicles. Applied energy, 88(3), 799-804.], [Song, Z., Zhang, X., Li, J., Hofmann, H., Ouyang, M., & Du, J. (2018). Component sizing optimization of plug-in hybrid electric vehicles with the hybrid energy storage system. Energy, 144, 393-403.], [Shivappriya, S. N., Karthikeyan, S., Prabu, S., Prado, R. P. D., & Parameshachari, B. D. (2020). A modified ABC-SQP-based combined approach for the optimization of a parallel hybrid electric vehicle. Energies, 13(17), 4529.], [Tran, M. K., Akinsanya, M., Panchal, S., Fraser, R., & Fowler, M. (2020). Design of a hybrid electric vehicle powertrain for performance optimization considering various powertrain components and configurations. Vehicles, 3(1), 20-32.]);
3) the unit of measure of variables are not always indicated in the text;
4) The quality and resolution of all the figures should be enhanced;
5) Section 2 and Section 3 are not readable. It is hard to understand all the reported steps since there are no comments that explain them.
6) Please, revise the optimization problem formulated in section 3 in order to improve it. As an example, see [Caiazzo, B., Coppola, A., Petrillo, A., & Santini, S. (2021). Distributed nonlinear model predictive control for connected autonomous electric vehicles platoon with distance-dependent air drag formulation. Energies, 14(16), 5122.].
7) In the reviewer's opinion, the main issue is related to contributions, which are unclear. For instance, in the introduction section the contributions, as well as a comparison with existing approaches, is not clearly highlighted.
8 ) A comparison with existing approaches will help to improve the analysis section. Comment 7 could help.
Author Response
To, Date: 01.12.2022
Editor-in-Chief
World Electric Vehicle Journal (WEVJ)
MDPI
St. Alban-Anlage 66, 4052 Basel, Switzerland
Reference:
Manuscript Reference No.: wevj-2005928
Submission Title: Fuel Efficiency Improvement by Component Size Optimization in Hybrid
Electric Vehicle
Manuscript Type: Research Article
Date of Submission: 17.10.2022
Subjects: Responses to Reviewers Comments for the Manuscript Reference No.: wevj-2005928
Dear Sir/Madam,
In view of above-mentioned reference, we express our sincere gratitude and thanks to Editor-in-Chief for accepting manuscript for submission and review process, we also express our gratitude and thanks to the eminent reviewers of the submitted manuscript for their review comments to revise the submitted manuscript for the next phase of publication.
Hereby we are submitting the responses of each review comments received through eminent reviewers. Appropriate changes related to review comments are incorporated in the revised manuscript. The responses as authors’ comments are mentioned below in the rebuttal table as follows.
Reviewer’s comments: |
Authors’ comments |
|
Review point: 1. |
The article is not easy to read and needs a major English-style update; |
Apologies for mistake, the correction is been done in revised manuscript. |
Review point: 2. |
The state of the art needs to be enhanced. For instance see ([Wu, X., Cao, B., Li, X., Xu, J., & Ren, X. (2011). Component sizing optimization of plug-in hybrid electric vehicles. Applied energy, 88(3), 799-804.], [Song, Z., Zhang, X., Li, J., Hofmann, H., Ouyang, M., & Du, J. (2018). Component sizing optimization of plug-in hybrid electric vehicles with the hybrid energy storage system. Energy, 144, 393-403.], [Shivappriya, S. N., Karthikeyan, S., Prabu, S., Prado, R. P. D., & Parameshachari, B. D. (2020). A modified ABC-SQP-based combined approach for the optimization of a parallel hybrid electric vehicle. Energies, 13(17), 4529.], [Tran, M. K., Akinsanya, M., Panchal, S., Fraser, R., & Fowler, M. (2020). Design of a hybrid electric vehicle powertrain for performance optimization considering various powertrain components and configurations. Vehicles, 3(1), 20-32.]); |
The literature review is upated with latest work in the area, The newly added references in literature review are [9], [12], [13], [16], [17], [18],[19], [20], [24] and [34] |
Review point: 3. |
the unit of measure of variables are not always indicated in the text; |
The required correction is been done in revised manuscript |
Review point: 4. |
The quality and resolution of all the figures should be enhanced; |
The image quality is improved in revised manuscript |
Review point: 5. |
Section 2 and Section 3 are not readable. It is hard to understand all the reported steps since there are no comments that explain them. |
The author has added new section IV to make the steps clear |
Review point: 6. |
Please, revise the optimization problem formulated in section 3 in order to improve it. As an example, see [Caiazzo, B., Coppola, A., Petrillo, A., & Santini, S. (2021). Distributed nonlinear model predictive control for connected autonomous electric vehicles platoon with distance-dependent air drag formulation. Energies, 14(16), 5122.]. |
The required correction is been done in revised manuscript |
Review point: 7. |
In the reviewer's opinion, the main issue is related to contributions, which are unclear. For instance, in the introduction section the contributions, as well as a comparison with existing approaches, is not clearly highlighted. |
The main contribution is mentioned in revised manuscript |
Review point: 8. |
A comparison with existing approaches will help to improve the analysis section. Comment 7 could help. |
The required correction is been done in revised manuscript |
The revised manuscript is also uploaded on Journal Management System portal.
Thanks & Regards
First Author:
Swapnil Srivastava
Department of Electrical Engineering
United College of Engineering and Research, Prayagraj,UP, India
Email id: [email protected]
Mobile/Phone: +91-8299652473, 9455419395
Corresponding author:
First Co-Author:
Sanjay Kumar Maurya, Ph.D.
SMIEEE (U.S.A.), MIET (U.K.)
Department of Electrical Engineering
GLA University, Mathura, UP, India
Email id: [email protected]
Phone No. +91-5446250855
Mobile-+91-7895010874
Reviewer 4 Report
The paper presented a DIRECT optimization algorithm for component sizing of HEV. 296
The proposed method is tested for various standard drive cycles including Indian scenario 297
drive cycles. The paper is interested and it's idea is suitable for publication. However, I have some comments.
-the main contributions of the paper should be clearly mentioned in the last phragrap in introduction
-the future work should be mentioned in conclusion
-please carefully check equation 47
Author Response
To, Date: 01.12.2022
Editor-in-Chief
World Electric Vehicle Journal (WEVJ)
MDPI
St. Alban-Anlage 66, 4052 Basel, Switzerland
Reference:
Manuscript Reference No.: wevj-2005928
Submission Title: Fuel Efficiency Improvement by Component Size Optimization in Hybrid
Electric Vehicle
Manuscript Type: Research Article
Date of Submission: 17.10.2022
Subjects: Responses to Reviewers Comments for the Manuscript Reference No.: wevj-2005928
Dear Sir/Madam,
In view of above-mentioned reference, we express our sincere gratitude and thanks to Editor-in-Chief for accepting manuscript for submission and review process, we also express our gratitude and thanks to the eminent reviewers of the submitted manuscript for their review comments to revise the submitted manuscript for the next phase of publication.
Hereby we are submitting the responses of each review comments received through eminent reviewers. Appropriate changes related to review comments are incorporated in the revised manuscript. The responses as authors’ comments are mentioned below in the rebuttal table as follows.
Reviewer’s comments: |
Authors’ comments |
|
Review point: 1. |
the main contributions of the paper should be clearly mentioned in the last phragrap in introduction |
The main contribution is added in introduction. |
Review point: 2. |
the future work should be mentioned in conclusion |
The future work is added in conclusion |
Review point: 3. |
please carefully check equation 47 |
The required corrections are made in revised manuscript |
The revised manuscript is also uploaded on Journal Management System portal.
Thanks & Regards
First Author:
Swapnil Srivastava
Department of Electrical Engineering
United College of Engineering and Research, Prayagraj,UP, India
Email id: [email protected]
Mobile/Phone: +91-8299652473, 9455419395
Corresponding author:
First Co-Author:
Sanjay Kumar Maurya, Ph.D.
SMIEEE (U.S.A.), MIET (U.K.)
Department of Electrical Engineering
GLA University, Mathura, UP, India
Email id: [email protected]
Phone No. +91-5446250855
Mobile-+91-7895010874
Round 2
Reviewer 1 Report
What is the relationship between the design variable parameters (such as cs_lo_SOC) and the component size (such as Energy Storage (kW))? It seems that the variable parameters can directly influence the energy consumption of vehicle, but the relationship between the component size and the energy consumption is not clearly described by equations.
Author Response
To, Date: 20.12.2022
Editor-in-Chief
World Electric Vehicle Journal (WEVJ)
MDPI
St. Alban-Anlage 66, 4052 Basel, Switzerland
Reference:
Manuscript Reference No.: wevj-2005928
Submission Title: Fuel Efficiency Improvement by Component Size Optimization in Hybrid
Electric Vehicle
Manuscript Type: Research Article
Date of Submission: 17.10.2022
Subjects: Responses to Reviewers Comments for the Manuscript Reference No.: wevj-2005928
Dear Sir/Madam,
In view of above-mentioned reference, we express our sincere gratitude and thanks to Editor-in-Chief for accepting manuscript for submission and review process, we also express our gratitude and thanks to the eminent reviewers of the submitted manuscript for their review comments to re-revise the submitted manuscript for the next phase of publication.
Hereby we are submitting the responses of each review comments received through eminent reviewers. Appropriate changes related to review comments are incorporated in the revised manuscript. The responses as authors’ comments are mentioned below in the rebuttal table as follows.
Reviewer’s comments: |
Authors’ comments |
|
Review point: 1. |
What is the relationship between the design variable parameters (such as cs_lo_SOC) and the component size (such as Energy Storage (kW))? It seems that the variable parameters can directly influence the energy consumption of vehicle, but the relationship between the component size and the energy consumption is not clearly described by equations. |
The relationship between design variable and component size is explained in section 3 in revised draft of the manuscript. |
The revised manuscript is also uploaded on Journal Management System portal.
Thanks & Regards
First Author:
Swapnil Srivastava
Department of Electrical Engineering
United College of Engineering and Research, Prayagraj,UP, India
Email id: [email protected]
Mobile/Phone: +91-8299652473, 9455419395
Corresponding author:
First Co-Author:
Sanjay Kumar Maurya, Ph.D.
SMIEEE (U.S.A.), MIET (U.K.)
Department of Electrical Engineering
GLA University, Mathura, UP, India
Email id: [email protected]
Phone No. +91-5446250855
Mobile-+91-7895010874
Reviewer 3 Report
The authors made a significant effort to improve the paper, but in the reviewer's opinion, the article still has serious deficiencies.
1) Section 3 still requires a significant update. Please improve the description of the different steps of the optimization problem. Moreover, a more formal and elegant formulation for the optimization control problem is required. Please, see again ([Caiazzo, B., Coppola, A., Petrillo, A., & Santini, S. (2021). Distributed nonlinear model predictive control for connected autonomous electric vehicles platoon with distance-dependent air drag formulation. Energies, 14(16), 5122.])
2) insert the label for the x axe in figure 7
3) The contents of section 6 are insufficient. The results presented in the tables and figures are not sufficiently explained. For instance, the results in Figure 8 are explained with just one phrase. this is a rough way of presenting the results.
4) What do the values ​​in table 6 mean? How is fuel efficiency calculated? The authors should better explain the results in the text, as well as in the caption of the table. Moreover, the Authors should also include a column with the percentage difference.
5) Comments similar to Comment 4 hold for all the reported results.
Author Response
To, Date: 20.12.2022
Editor-in-Chief
World Electric Vehicle Journal (WEVJ)
MDPI
St. Alban-Anlage 66, 4052 Basel, Switzerland
Reference:
Manuscript Reference No.: wevj-2005928
Submission Title: Fuel Efficiency Improvement by Component Size Optimization in Hybrid
Electric Vehicle
Manuscript Type: Research Article
Date of Submission: 17.10.2022
Subjects: Responses to Reviewers Comments for the Manuscript Reference No.: wevj-2005928
Dear Sir/Madam,
In view of above-mentioned reference, we express our sincere gratitude and thanks to Editor-in-Chief for accepting manuscript for submission and review process, we also express our gratitude and thanks to the eminent reviewers of the submitted manuscript for their review comments to re-revise the submitted manuscript for the next phase of publication.
Hereby we are submitting the responses of each review comments received through eminent reviewers. Appropriate changes related to review comments are incorporated in the revised manuscript. The responses as authors’ comments are mentioned below in the rebuttal table as follows.
Reviewer’s comments: |
Authors’ comments |
|
Review point: 1. |
Section 3 still requires a significant update. Please improve the description of the different steps of the optimization problem. Moreover, a more formal and elegant formulation for the optimization control problem is required. Please, see again ([Caiazzo, B., Coppola, A., Petrillo, A., & Santini, S. (2021). Distributed nonlinear model predictive control for connected autonomous electric vehicles platoon with distance-dependent air drag formulation. Energies, 14(16), 5122.]) |
Section 3 is updated as per the valuable suggestion provided by the eminent reviewer. |
Review point: 2. |
insert the label for the x axe in figure 7 |
The label of x-axis in figure 7 is updated, apologies for mistake |
Review point: 3. |
The contents of section 6 are insufficient. The results presented in the tables and figures are not sufficiently explained. For instance, the results in Figure 8 are explained with just one phrase. this is a rough way of presenting the results. |
The result explanation is further elaborated. |
Review point: 4. |
What do the values ​​in table 6 mean? How is fuel efficiency calculated? The authors should better explain the results in the text, as well as in the caption of the table. Moreover, the Authors should also include a column with the percentage difference. |
Explanation of result is elaborated and a column to indicate the percentage difference is also added. |
Review point: 5. |
Comments similar to Comment 4 hold for all the reported results. |
Explanation of result is elaborated and a column to indicate the percentage difference is also added for other results. |
The revised manuscript is also uploaded on Journal Management System portal.
Thanks & Regards
First Author:
Swapnil Srivastava
Department of Electrical Engineering
United College of Engineering and Research, Prayagraj,UP, India
Email id: [email protected]
Mobile/Phone: +91-8299652473, 9455419395
Corresponding author:
First Co-Author:
Sanjay Kumar Maurya, Ph.D.
SMIEEE (U.S.A.), MIET (U.K.)
Department of Electrical Engineering
GLA University, Mathura, UP, India
Email id: [email protected]
Phone No. +91-5446250855
Mobile-+91-7895010874
Round 3
Reviewer 3 Report
The paper can be now published in its present form.